# Disengagement from treatment and its socio-demographic and clinical predictors among patients with incident schizophrenia in a Nigerian psychiatric hospital: 8-year naturalistic follow-up analysis

**Justus Uchenna Onu**[1,2]*, **Temitope Ibukun Olatayo**[2], **Obioma Comfort Okoye**[2], **Nneoma Chizaram Akujobi**[2], **Jude Uzoma Ohaeri**[3]

**1** Department of Mental Health, Nnamdi Azikiwe University, Awka, Anambra State, Nigeria, **2** Department of Training and Research, Federal Neuropsychiatric Hospital, Enugu, Nigeria, **3** Department of Psychological Medicine, University of Nigeria, Nsukka, Enugu State, Nigeria

* ju.onu@unizik.edu.ng

## Abstract

### Background

Schizophrenia is a chronic-relapsing condition that in most cases, requires life-long treatment and follow-up. However, disengagement from care threatens the achievement of therapeutic targets for these patients, their families and the society. This study aimed to determine the burden of disengagement, and the socio-demographic and clinical predictors among patients with incident schizophrenia in a Nigeria Psychiatric Hospital in 8 years of follow-up.

### Method

This was a naturalistic study involving 160 clinically well-characterized patients with schizophrenia on follow-up, whose outcome data for 16 weeks had been reported. Subsequent follow-up data during scheduled appointments were obtained directly from the patients, case notes and via telephone interviews with participants and/or their caregivers. Disengagement from care and time to disengagement were operationally defined. The proportion that disengaged was calculated using percentages and 95% confidence interval. Time to disengagement was estimated using Kaplan-Meier time-to-event analysis, while the associated factors were analyzed by logistic regression.

### Results

At the 8th year of follow-up, we had contact with 147 (91.9%) subjects, mostly by phone, out of whom 82.3% (95% CI: 75.2%–88.1%) had disengaged from care. The median time to disengagement and number of visits in 8 years were: 1 year and five visits, respectively. A quarter (40/160) of the original participants never came for their first appointment. The

**Data availability statement:** The raw data is available and submitted as supporting material.

**Funding:** The authors received no specific funding for this work.

**Competing interests:** The authors have declared that no competing interests exist.

common reasons given for disengagement were financial constraints and dissatisfaction with improvement. A quarter (38/147) had sought care with religious and traditional faith healers after encountering our service. The mortality rate was 47.6 per 1000 (7/147). Cause of death was attributed to suicide (3/7) and other chronic medical conditions (3/7). Those who relapsed had significantly longer time to disengagement than those without relapse.

## Conclusion

Disengagement is a common phenomenon in our setting, as elsewhere in the literature, despite adequate family support and good clinical outcome. The finding on the time to disengagement has implications for designing interventions to promote adherence to treatment plans.

## Introduction

Schizophrenia is a chronic-relapsing condition that in most cases, will require a lifelong follow-up treatment to minimize relapse and facilitate recovery [1]. Disengagement from mental healthcare services among persons with schizophrenia can alter the therapeutic targets with dire consequences to the individuals, their families and the society [2].

Though there is a lack of consensus on its definition, disengagement from treatment or attrition from follow-up care, whether manifested as non-attendance, non-completion of treatment goals or poor engagement with the healthcare personnel, is a common problem in general clinical settings [2,3]. One out of every three persons with severe mental illness (SMI) disengages from care, with even higher rates reported in many studies, depending on the operational definition of disengagement [4–6]. The rate of disengagement is highest at the point of initiation or transition of care, with one review indicating that one out of every two persons with SMI fails to attend their first follow-up visit [4]. Studies examining the time taken before disengagement from mental health care services have reported variable results depending on the population studied and the length of follow-up [7,8]. However, on the average, one recent meta-analysis reported a median time of 15 months with a range of 5–22.4 months [9].

Predictors of disengagement also vary across studies with only a few consistent findings, suggesting the complexity and the multifaceted nature of the phenomenon [4,7,8]. Commonly replicated predictors of disengagement among western populations are: young age, ethnic minorities and deprived populations, lack of insight, substance use disorders, forensic history, duration of untreated psychosis, severity of symptoms at baseline and service-level [8]. Of note, the contribution of mortality to disengagement has received scant attention; and this is important in our setting in sub-Saharan Africa (SSA), which records one of the highest mortality rates in the world [10].

Given the importance of continuity of care in schizophrenia and the evidence suggesting that long-term care can improve symptoms and functioning, as well as reduce relapse risk [1,11,12], disengagement from treatment is a serious concern for mental healthcare services. Despite the usefulness of treatment engagement, only a few studies have addressed the rates and predictors of disengagement from care among patients with schizophrenia in SSA using rigorous methodology. Based on the foregoing, the following questions become pertinent: what is the rate of disengagement among a cohort of incident cases of schizophrenia at 8-year naturalistic follow-up? What is the median time to disengagement from mental healthcare services among patients with incident cases of schizophrenia? What are the socio-demographic

and clinical predictors of disengagement from care among patients with schizophrenia? What is the contribution of mortality to this outcome? Does contact with orthodox mental health-care services terminate the pathway to care among patients with schizophrenia?

## Materials and methods

### Study design and population

This was a naturalistic longitudinal follow-up outcome study which took place at the Federal Neuropsychiatric Hospital (FNH), Enugu, Nigeria. Details of the study methodology have been described in two previous reports from this cohort [13,14]. Baseline recruitment was done from April to July 2016. During the recruitment period, consecutive incident cases of schizophrenia, who presented at the hospital, aged 18–49 years, had traceable home address around Enugu metropolis, as well as mobile telephones, were included in the study. It is usual for patients with severe mental disorders (e.g., schizophrenia and bipolar disorders) to be accompanied to the hospital by at least one family member, even if the patient is in remission. Nigerian psychiatrists encourage this practice in order to obtain more detailed clinical and relevant social information. The well-known extended family system in the culture supports the ready availability of this social support.

Capacity to participate was determined by the research team and an independent psychiatrist, based on the following criteria: (1) if the participant understood and communicated the purpose of the study, (2) understood and communicated the study procedure; and (3) understood the risks and benefits for participating.

### Operational definition of outcomes measures

Although our definition of treatment disengagement included: not taking the prescribed medications, not attending scheduled clinic appointment in the past 12 months and leaving the hospital against medical advice [15], our ultimate dependent variable was the time to loss to follow-up. We excluded participants whose case notes were not traceable, those appropriately referred out by the managing team, and those whose treatment were discontinued by the clinicians. Based on the above exclusion criteria, we excluded 12 participants whose case notes were not traceable and one who was referred out to a neighboring facility for continuation of care. Time to disengagement (i.e., survival time) was defined as the time in years from the baseline assessment to the time the patient was lost to follow-up [15].

### Procedure

The summary of methodological considerations with regards to the index report [13,14], is described herein. At the baseline, majority of the new cases of schizophrenia were offered hospitalization, and their initial assessments were completed in the wards after obtaining a written informed consent. Diagnosis was based on the International Classification of Mental and Behavioral Disorders (ICD-10).

First, using the screening section of the modules of the MINI, the authors sought to screen out the presence of co-morbid major mental disorders. Thereafter, the criteria for schizophrenia were confirmed using the Mini International Neuropsychiatric Interview (MINI) schizophrenia (psychotic disorders) module. For individuals who neither met the diagnostic criteria nor gave consent to participate in the study, records of their diagnoses and reasons for refusal were documented. After a detailed medical history, a trained research assistant did a full physical examination (including neurological examination) to exclude the presence of co-morbid physical conditions. We then administered the socio-demographic questionnaire; this

questionnaire contained items to assess socio-demographic characteristics and other clinical variables, such as duration of untreated psychosis (defined as the time from onset of psychosis to onset of treatment, i.e., at baseline evaluation, as highlighted in the earlier reports) [13,14].

The baseline clinical profile was assessed, using the Brief Psychiatric Rating Scale (BPRS), the Scale for Assessment of Negative Symptoms (SANS), and the CGI (severity). Then, the baseline psychosocial functioning of the participants was assessed using the World Health Organization Disability Assessment Scale (WHODAS) and the American Psychiatric Association's Global Assessment of Functioning scale (GAF). Also, the social support scale, i.e., the Multi-dimensional Scale of Perceived Social Support (MSPSS) was applied when the patient was judged to have become clinically stable and able to complete the questionnaire.

After baseline assessment, the initial follow-up was done 4-weekly up till 16 weeks, and subsequently yearly using physical contacts during scheduled visits, phone calls for those not attending and the case files for those who dropped in on unscheduled days at the emergency department. The current report is focused on the 8th year of follow-up, and the final assessments were done from January to May, 2024. Data collection for this report was by physical contact for those who attended scheduled outpatient appointments, phone calls for participants (and family caregivers) who defaulted appointments but could be reached by phone, and complemented by information from the case notes. The choice of this method was based on the following considerations: (1) there was the need to speak with several family caregivers to obtain corroborative information of interest to the researchers; (2) the case notes provided multiple information, including the details of participants' clinical visits (i.e., time when the last visit was made, number of visits, number of admissions and relapses etc.); (3) the cost of transportation and the risk of road travel in the study setting was a concern to the researchers due to worsening security challenges. We defined symptomatic remission as a rating of 'mild' or less, concurrently on the following seven BPRS items (BPRS only criteria): grandiosity, suspiciousness, unusual thought content, hallucinatory behavior, conceptual disorganization, mannerism/posturing, and blunted affect [14].

## Cohort register

At the initial phase of the study, the authors established a register for this cohort. This register contained the socio-demographic characteristics of the participants, hospital numbers, telephone numbers of patients, multiple family members and caregivers and house/home address. The authors have also updated the register based on changes in phone numbers and addresses to aid follow-up. Participants who died were documented. The baseline socio-demographic characteristics such as age, gender, living circumstances and social support variables were documented. Similarly, the clinical variables such as age at onset, duration of illness, family history of mental illness and type of medication were also documented.

## Data extraction

The following variables were of interest: employment status, number of relapses in the past 8 years, number of admissions, engagement with treatment, year the participant was lost to follow-up, time it took from baseline to disengagement, in years, whether treatment was sought elsewhere, other places where treatment was sought, whether participant was alive or dead, (and if dead, the family or caregiver's understanding of the cause of death), if participant was still having symptoms, medication status and the reasons for disengaging from care.

We reviewed the case notes of all 160 participants included in the baseline study at each scheduled follow-up visit, during which patients were interviewed by members of the research team. TIO, OCO, NCA and JUO carefully extracted all relevant data from the case notes and

thereafter interacted (i.e., 90 participants could be reached via telephone) with the patients and the family members, to ascertain the information highlighted above. The intervals of scheduled clinic appointments were naturalistic, i.e., based on the patient's clinical situation, and ranged from monthly to 3-monthly. At intervals of follow-up, patients who missed their appointment were interviewed by telephone for the above information and to inquire into reasons for disengagement for those lost to follow-up. In case of death, information obtained from family members by telephone were: exact date of demise, possible causes of mortality and the medication status at the time of death.

## Ethical considerations

Ethical approval was obtained from the Ethics and Research Committees of the institution with approval number of FNHE/HCS&T/REA/Vol.1/187. International ethical norms and standards as stipulated in the Helsinki guideline were strictly adhered to; written informed consent was obtained from all the participants. Participants were free to withdraw from the study at any time, even after having consented initially, and this did not in any way affect the patient's medical care or their academic endeavors. They were also informed that if in the process of the research, any previously undiagnosed clinical condition was found, the managing team would be appropriately informed. In addition, participants were told that answering the research questions does not replace their routine follow-up visit. Information gathered in the course of the research was stored in an encrypted computer accessible only to the researcher.

## Data analysis

Data was analyzed using IBM-SPSS version 20. The normality of distribution of continuous variables were checked using Shapiro-Wilk test. Categorical variables (such as engagement status) were described using frequency counts and percentages, while continuous variables (such as number of follow-up visits) were described using median and interquartile range. The mortality rate was calculated per 1000 people. Multivariate logistic regression analysis was used to determine the predictors of disengagement from treatment. The Kaplan-Meier survival analysis with Log Rank test was used to check statistical differences in the time to the event (i.e., disengagement) for socio-demographic and clinical variables. In computing the Kaplan-Meier analyses, we employed the independence of censoring and events. Participants who were ascertained to have disengaged from care were recorded as the event and coded as 1, while those who were still engaged with care or had left for reasons not related to disengagement (e.g., official referral to other centers, and death), were censored and coded as 0. All tests were two-tailed at 95% confidence interval and p-value was considered significant if <0.05.

## Results

### Socio-demographic and clinical profile of the participants

The participants were mostly young, 18–49 years, mean age of 31 (SD 7.8) year, and 48.8% were males (Table 1). They were mostly single (66.3%), unemployed (63.8%), had high school level of education (74.4%) and lived under the care of family members (69.4%). Table 2 shows the baseline clinical characteristics of the participants. The median age at onset of illness and duration of illness at entry into the study were 25 years and 32 months, respectively. The mode of onset of illness was insidious in 58.1%, and majority were on haloperidol (48.8%) and olanzapine (24.4%).

The 8-year follow-up characteristics are shown in Table 3 and Fig 1.

**Table 1. Baseline socio-demographic characteristics of the participants. N = 147.**

| Characteristics | Frequency (%) | Mean ± SD |
|---|---|---|
| **Age of participants (years)** | | 30.78 ± 7.81 |
| **Gender** | | |
| Male | 72 (49.0) | |
| Female | 75 (51.0) | |
| **Marital status** | | |
| Living with a partner | 39 (26.5) | |
| Not living with a partner | 108 (73.5) | |
| **Level of education** | | |
| None/≤6 years | 35 (23.8) | |
| >6 years | 112 (76.2) | |
| **Employment status** | | |
| Unemployed | 93 (63.3) | |
| Employed | 54 (36.7) | |
| **Care giver relationships** | | |
| Living alone | 13 (8.8) | |
| Living with others | 134 (91.2) | |
| **Perceived social support** | | |
| Satisfactory | 41 (31.1) | |
| Unsatisfactory | 106 (72.1) | |

SD = Standard Deviation

**Table 2. Baseline clinical characteristics of the 147 participants.**

| Characteristics | Frequency (%) | Median (IQR) |
|---|---|---|
| **Age at onset (years)** | | 15.0 (11.00) |
| **Duration of illness in months** | | 32.00 (85.00) |
| ≥5 years | 55 (37.4) | |
| <5 years | 92 (62.6) | |
| **Mode of onset** | | |
| Acute | 60 (40.8) | |
| Insidious | 87 (59.2) | |
| **Family history of mental illness** | | |
| Positive | 85 (57.8) | |
| Negative | 62 (42.2) | |
| Systolic blood pressure (mmHg) | | 110.00 (20.00) |
| Diastolic blood pressure (mmHg) | | 60.00 (20.00) |
| Waist circumference (cm) | | 82.00 (17.00) |
| Hip circumference (cm) | | 91.00 (10.00) |
| Waist-Hip ratio | | 0.85 (0.05) |

IQR = Interquartile Range

At the 8th year of follow-up, we contacted 147 (91.9%) of the original 160 participants and their family members, either physically, or by telephone, with information supplemented from the case records. Of this number, 121(82.3%, 95% C.I., 75.2%–88.1%) had disengaged, and

**Table 3. 8-Year follow-up characteristics of disengagement variables of the participants.**

| Variables | Frequency (%) |
|---|---|
| **Disengagement from care (n = 147)** | |
| Disengaged | 121 (82.3) [95% CI, 75.2%–88.1%] |
| Engaged | 26 (17.1) |
| **Median time to disengagement in years (IQR)** | 1.00 (2.50) |
| **Number of visits in eight years (n = 147)** | |
| None after initial intake | 36 (24.5) |
| 1–5 visits | 49 (33.3) |
| >5 visits | 62 (42.2) |
| **Median number of visits (IQR)** | 5.00 (14.00) |
| **Mortality status (n = 147)** | |
| Death | 7 (4.8) |
| Missing | 6 (4.1) |
| **Mortality rate per 1000 population** | 47.6 |
| **Verbally reported cause of death (n = 7)** | |
| Suicide | 3 (42.9) |
| Other health conditions | 3 (42.9) |
| Accidental | 1 (14.3) |
| **History of relapse (n = 147)** | |
| Yes | 71 (48.3) |
| No | 76 (51.7) |
| **Number of relapse (n = 71)** | |
| Once | 27 (38.0) |
| 2–4 | 38 (53.5) |
| ≥5 | 6 (8.5) |
| **Currently on medication (n = 147)** | |
| Yes | 23 (15.6) |
| No | 50 (34.0) |
| Not sure* | 74 (50.4) |
| **Self/caregiver-reported current symptoms (n = 147)** | |
| Present | 40 (27.2) |
| Absent | 30 (20.4) |
| Not sure* | 77 (52.4) |
| **Self/caregiver-reported functioning (n = 147)** | |
| Currently engaged in occupational activities | 34 (23.1) |
| Not working | 37 (25.2) |
| Not sure* | 76 (51.7) |
| **Sought alternative treatment (n = 147)** | |
| Yes | 38 (25.9) |
| No | 33 (22.4) |
| Not sure* | 76(51.7) |
| **Place where alternative treatment was sought (n = 38)** | |
| Traditional/herbalist | 8 (21.1) |
| Religious centers | 27 (71.1) |
| Combining orthodox and unorthodox care | 3 (7.8) |

*Caregiver interviewed and was not sure of the current status of the patient, or not available in case file.

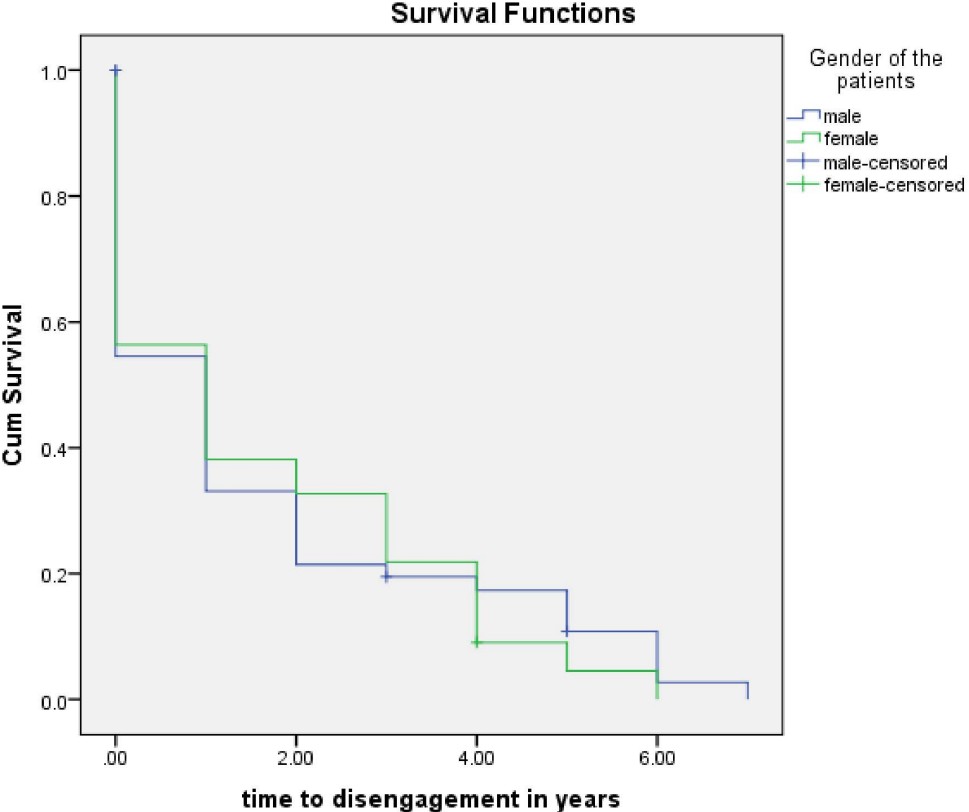

**Fig 1. Time to disengagement between the genders.**

Kaplan–Meir survival analysis indicated that the median time to disengagement was 1 year for both genders, as shown in Fig 1.

About a quarter of the 147 subjects (36/147) made no visits after intake, while 42.2% (62/147) made over 5 visits during the period. Other follow-up characteristics as shown in Table 3 are as follows: 15.6% (23/147) were still taking their medication, 27.2% (40/147) had significant symptoms, 23.1% (34/147) were engaged in occupational activities, and 25.8% (38/147) had sought treatment elsewhere. Of the 38 subjects who indicated they had sought treatment elsewhere after contacting our service, most of them (71.1% – 28/38) patronized Christian religious faith healers.

Associations of time to disengagement: Table 4.

Our Kaplan-Meier analysis assessed whether any of the categories of socio-demographics (including gender and perceived social support), clinical (e.g., mode of onset, duration of illness) and clinical outcome (at weeks 4, 8, 12 and 16) characteristics was associated with a significantly longer time to disengagement. Although there were no significant associations for most variables (p > 0.05), there was a tendency for those with optimal remission (versus those without remission) at 12th and 16th weeks, to have a shorter time to disengagement (1.37 years vs., 2.28 years, p = 0.06; and 1.29 vs., 2.25 years, p = 0.07, respectively). Accordingly, time to disengagement was significantly shorter for those who did not have relapses (0.92 years), vs., those who had relapses (2.22 years) (p < 0.01)

Furthermore, Table 5 shows that, in logistic regression analysis, with categories of baseline socio-demographics and clinical characteristics (including functional disability and

Table 4.  Time to Disengagement vs. independent variables. N = 147.

| Independent variables | Time to disengagement mean (S.E) | 95% CI | p-value* |
|---|---|---|---|
| **Marital status** | | | 0.66 |
| Living with a partner | 1.66 (0.36) | 0.94–2.38 | |
| Not living with a partner | 1.91 (0.39) | 1.13–2.68 | |
| **Education status** | | | 0.37 |
| None/≤6 years | 1.39 (0.37) | 0.65–2.12 | |
| >6 years | 2.04 (0.41) | 1.23–2.84 | |
| **Living status** | | | 0.23 |
| Lives alone | 1.20 (0.42) | 0.37–2.04 | |
| Lives with relatives | 1.96 (0.35) | 1.27–2.64 | |
| **Perceived social support** | | | 0.31 |
| Satisfactory | 2.52 (0.57) | 1.40–3.65 | |
| Unsatisfactory | 2.05 (0.48) | 1.10–2.99 | |
| **Mode of onset** | | | 0.77 |
| Acute | 1.73 (0.33) | 1.07–2.39 | |
| Insidious | 1.86 (0.41) | 1.04–2.67 | |
| **Duration of illness** | | | 0.85 |
| ≤ 5 years | 1.65 (0.23) | 1.19–2.11 | |
| >5 years | 2.08 (0.68) | 0.68–3.42 | |
| **Remission at 4th week** | | | 0.66 |
| Yes | 2.03 (0.40) | 1.28–2.82 | |
| No | 1.76 (0.25) | 1.26–2.26 | |
| **Remission at 8th week** | | | 0.18 |
| Yes | 2.19 (0.32) | 1.55–2.83 | |
| No | 1.48 (0.28) | 0.91–2.04 | |
| **Remission at 12th week** | | | 0.06 |
| Yes | 2.28 (0.34) | 1.60–2.96 | |
| No | 1.37 (0.29) | 0.79–1.95 | |
| **Remission at 16th week** | | | 0.07 |
| Yes | 2.25 (0.31) | 1.62–2.87 | |
| No | 1.29 (0.33) | 0.64–1.99 | |
| **Relapse in 8 years** | | | 0.01** |
| Yes | 2.22 (0.33) | 1.56–2.88 | |
| No | 0.92 (0.26) | 0.40–1.45 | |

SE = Standard Error, CI = Confidence Interval, * = Log Rank, ** = Significant p-value.

psychopathology score) as independent variables, there were no significant baseline predictors of time to disengagement.

## Discussion

The study aimed to determine the rate and time of disengagement from 8-year follow-up care and its socio-demographic and clinical predictors for 160 incident cases of schizophrenia. There is a dearth of long-term longitudinal outcome studies in Africa in general [16,17], and Nigeria in particular [18]. Furthermore, the authors are not aware of studies in Africa focusing on disengagement from follow-up care in the long-term. The major strength of our study is

**Table 5.  Baseline predictors of disengagement from care at 8 years of follow-up. N = 147.**

| Baseline variables | Wald | AOR | 95% CI | p-value |
|---|---|---|---|---|
| **Age** | 0.4 | 0.79 | 0.39–1.61 | 0.52 |
| **Sex** | | | | |
| Male | 1.78 | 2.32 | 0.67–8.09 | 0.18 |
| Female | 1 | | | |
| **Marital status** | | | | |
| Not living with a partner | 2.85 | 0.29 | 0.06–1.22 | 0.09 |
| Living with a partner | 1 | | | |
| **Education** | | | | |
| ≤6 years of education | 1.27 | 2.25 | 0.55–9.24 | 0.25 |
| >6 years of education | 1 | | | |
| **Living status** | | | | |
| Living alone | 1.27 | 0.37 | 0.06–2.05 | 0.37 |
| Living with relatives | 1 | | | |
| **Social support** | | | | |
| Unsatisfactory | 0.01 | 0.95 | 0.26–3.45 | 0.93 |
| Satisfactory | 1 | | | |
| **Mode of onset** | | | | |
| Insidious | 2.28 | 2.42 | 0.76–7.62 | 0.13 |
| Acute | 1 | | | |
| Baseline GAF score | 0 | 1 | 0.93–1.06 | 0.99 |
| Baseline WHODAS score | 0.02 | 0.99 | 0.95–1.04 | 0.87 |
| Baseline BPRS score | 0.29 | 0.98 | 0.92–1.04 | 0.58 |
| Age at onset (years) | 0.28 | 1.2 | 0.96–1.08 | 0.59 |
| Duration of illness at baseline (months) | 0.67 | 1.02 | 0.96–1.08 | 0.41 |
| GAF score at 4th week | 1.19 | 1.07 | 0.96–1.21 | 0.27 |
| WHODAS score at 4th week | 0.57 | 1.02 | 0.96–1.09 | 0.57 |
| BPRS score at 4th week | 0.09 | 0.98 | 0.88–1.09 | 0.76 |
| GAF score at 8th week | 0.22 | 0.96 | 0.83–1.12 | 0.63 |
| WHODAS score at 8th week | 0.29 | 1.02 | 0.91–1.15 | 0.59 |
| BPRS score at 8th week | 1.81 | 1.1 | 0.95–1.28 | 0.44 |
| GAF score at 12th week | 0.26 | 1.02 | 0.90–1.18 | 0.6 |
| WHODAS score at 12th week | 0.06 | 1.02 | 0.92–1.11 | 0.79 |
| BPRS score at 12th week | 0.4 | 1.06 | 0.88–1.02 | 0.97 |
| GAF score at 16th week | 0.1 | 1.01 | 0.91–1.13 | 0.74 |
| WHODAS score at 16th week | 1.98 | 1.07 | 0.97–1.19 | 0.15 |
| BPRS score at 16th week | 2.77 | 0.8 | 0.62–1.11 | 0.09 |

Dependent variable = Disengagement status (engaged/disengaged), AOR = Adjusted Odds Ratio, CI = Confidence Interval, GAF = Global Assessment of Functioning, WHODAS = World Health Organization's Disability Assessment Scale, BPRS = Brief Psychiatric Rating Scale.

that incident cases, predominantly neuroleptic naïve at the baseline, were followed up for a relatively long time, thus overcoming some of the methodological issues in previous studies [16,17].

The highlights of the findings are: (1) a quarter of the participants never attended scheduled first appointment; (2) eight out of every 10 participants at baseline had disengaged from treatment at 8 years, with the median time to disengagement of 1 year; (3) although none of

the categories of socio-demographic, clinical and treatment outcome characteristics at baseline was associated with longer time to disengagement, those who had relapses took significantly longer time to disengage, vs., those without relapses; and (4) about a quarter of the 147 who could be contacted had sought treatment at religious and traditional faith healing centers after contacting our service.

The finding that about a quarter of the participants did not attend their first appointment after the baseline visit fell within the range of 18%–67% reported in previous studies [2,4,19]. This is also in consonance with the observation of O'Brien et al [4]; that the initial period of treatment is the most likely time for dropout to occur. Our prevalence of disengagement (82%) was also similar to reports from studies with similar follow-up period (71.0%–88.1%) [2,15,19], while being much higher than others with shorter follow-up period [4,5].This finding agrees with the reports in the literature that failure to adhere to treatment program is a major issue in the management of patients with psychotic illnesses [4–7]. The variations in the disengagement rates reported across studies may be related to diverse factors. First, the length of follow-up is an important variable. A review of naturalistic studies showed that by the 6th month of treatment, 33% to 44% of patients had dropped out, and, by 1 year, 59% had dropped out [20–22]. Second, the operational definition of disengagement is another important consideration in explaining the variations in the rates across studies. For example, authors who defined disengagement as "no contact with mental health services for a continuous period of 3 months in a 2-year follow-up program", reported a rate of 28% [7]; those who defined it as "terminating treatment against medical advice", reported 34% [23]; and it was 50% for one study that used an engagement scale [24]. Third, is the population studied or the study setting. There is some evidence that disengagement rate is higher in populations with indices of social disadvantage or ethnic minorities [25]. This is particularly important in our study setting where the health system is weak, with no health insurance coverage for the unemployed and also for most of those employed, who constitute the majority of our patients [26]. Hence, our finding of high disengagement rate may be related to finance. Indeed, majority of the participants and family members cited economic difficulties as the major reason for disengagement from care.

Regarding the time to disengagement, we found a median time of 1 year. This is consistent with a recent meta-analysis that included nine studies that reported on time to disengagement among eight cohorts over 2–3 years. The authors found time to disengagement ranging from 5 months to 22.4 months with a median time of 15 months [9]. Five of the studies in the meta-analysis utilized Kaplan-Meier Time-to-Event analysis which is the methodology employed in the index study. This finding should inform the timing of interventions to sustain engagement in mental health care services.

The baseline socio-demographic and clinical characteristics did not significantly predict disengagement at the 8th year in the index study. Previous studies have given inconsistent evidence with regards to the baseline predictors of disengagement [4,7,8]. The most replicated predictors of disengagement in the literature are: baseline and persistent substance abuse, duration of untreated psychosis and living alone or with no family [9]. Other variables, such as age, gender, symptom dimensions, baseline functioning, and family history of schizophrenia had inconsistent results, with most studies reporting non-significant associations and a few finding significant associations [9]. The variability of these factors across studies, with only few consistent associations, suggests that disengagement from treatment in mental health-care settings is a complex and multifaceted phenomenon. Important is the consideration of other factors that may be more pertinent to engagement with care in the study setting, such as access to the follow-up appointments, the role of faith/religious healing centers in moving patients away from formal care services, and the local socio-cultural explanatory models of

mental disorders. These factors may be more relevant in our setting than the variables in previous studies. Such factors are important because they can be addressed directly by the service systems.

Another important finding in the index study is that a quarter of the participants sought alternative care in faith-based institutions even after coming in contact with mental healthcare specialists. In SSA, the pathway to mental healthcare services has been fairly described in previous studies [27–29]. A recent review of studies of pathway to care in SSA indicated that there was equal initial choice of biomedical (49.2%) and alternative (48.1%) mental healthcare pathways [29]. In the review cited above, it appears that regional differences were evident, with Western and Eastern Africa having preference for traditional care, while Southern Africa tilted towards biomedical care [29]. However, a fundamental question that has not been evaluated to the authors' knowledge in the region is: whether the pathway to care ends with coming in contact with orthodox mental healthcare services? From our findings, it appears that the pathway to mental healthcare services in Nigeria is complex; with some remaining in orthodox care, while about a quarter returned to the earlier path taken (i.e., faith-based care) and a few utilized both orthodox and alternative care simultaneously. A number of factors may be responsible for this finding: First, the ease of availability and the perceived cheaper cost of care in the faith-based institutions may be a factor. With poor health insurance coverage and mostly out-of-pocket expenses in Nigeria, it is not surprising that faith-based care will be sought. Second, the perceived efficacy and religious affiliations enhance its acceptability in the culture. Third, it is possible that the popularity of the supernatural explanatory model of mental illness in Africa has influenced the help seeking behavior of this cohort. This finding has implications for designing interventions to improve retention, using appropriate culturally informed strategies.

## Limitations

One of the limitations of this study was the issue of missing folders. Although this is expected in a naturalistic study, we have tried to control for this limitation by restricting the paper to only the 147 participants who could be reached, and described some characteristics of the 13 who could not be contacted. The second limitation was the inability to follow-up patients at their homes, because we had no funding for the work

## Conclusion

Disengagement from mental healthcare services in our SSA setting is as common in the treatment journey as elsewhere reported in the literature. The 1-year median time to disengagement, similar to the findings in the literature, should inform the timing of interventions to improve engagement with mental healthcare services. The finding on patronage of faith-based care should inform public mental health education.

## Supporting information

**S1 Data:** Dataset for the 8-year follow-up of the cohort(SAV)

## Author contributions

**Conceptualization:** Justus Uchenna Onu, Temitope Ibukun Olatayo, Obioma Comfort Okoye, Nneoma Chizaram Akujobi, Jude Uzoma Ohaeri.

**Data curation:** Justus Uchenna Onu, Temitope Ibukun Olatayo, Obioma Comfort Okoye, Nneoma Chizaram Akujobi.

**Formal analysis:** Justus Uchenna Onu, Jude Uzoma Ohaeri.

**Methodology:** Justus Uchenna Onu, Temitope Ibukun Olatayo, Obioma Comfort Okoye, Nneoma Chizaram Akujobi, Jude Uzoma Ohaeri.

**Supervision:** Jude Uzoma Ohaeri.

**Writing – original draft:** Justus Uchenna Onu.

**Writing – review & editing:** Justus Uchenna Onu, Temitope Ibukun Olatayo, Obioma Comfort Okoye, Nneoma Chizaram Akujobi, Jude Uzoma Ohaeri.

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
