## [Decision Letter · Decision Letter 0]

25 Oct 2024

PMEN-D-24-00308

Disengagement from Treatment and its Socio-demographic and Clinical Predictors among Patients with Incident Schizophrenia in a Nigerian Psychiatric Hospital: 8-Year Naturalistic Follow-up Analysis

PLOS Mental Health

Dear Dr. Onu,

Thank you for submitting your manuscript to PLOS Mental Health. After careful consideration, we feel that it has merit but does not fully meet PLOS Mental Health’s publication criteria as it currently stands. Therefore, we invite you to submit a revised version of the manuscript that addresses the points raised during the review process.

The manuscript has been evaluated by three reviewers, and their comments are available below.

The reviewers have raised a number of concerns that need attention. They request improvements to the reporting of methodological aspects of the study, and revisions to the interpretation of the results, and subsequent discussion.

Could you please revise the manuscript to carefully address the concerns raised?

We look forward to receiving your revised manuscript.

Kind regards,

Helen Howard

Staff Editor

PLOS Mental Health

Journal Requirements:

Additional Editor Comments (if provided):

Reviewers' comments:

Reviewer's Responses to Questions

**Comments to the Author**

1. Does this manuscript meet PLOS Mental Health’s publication criteria?

Reviewer #1: Partly

Reviewer #2: Partly

Reviewer #3: Yes

2. Has the statistical analysis been performed appropriately and rigorously?

Reviewer #1: No

Reviewer #2: Yes

Reviewer #3: Yes

3. Have the authors made all data underlying the findings in their manuscript fully available (please refer to the Data Availability Statement at the start of the manuscript PDF file)?

Reviewer #1: Yes

Reviewer #2: Yes

Reviewer #3: Yes

4. Is the manuscript presented in an intelligible fashion and written in standard English?

Reviewer #1: Yes

Reviewer #2: No

Reviewer #3: Yes

Reviewer #1: Examining the time to disengagement with mental health services among people with schizophrenia is a very good idea, and useful to have eight years' duration. There are however some areas where the methods and results are not clear, and greater precision in reporting would strengthen the paper. For example:

- Study design is unclear

A "naturalistic longitudinal follow-up outcome study" suggests a prospective observational study. However, on further reading it appears that the study is a retrospective record review of a cohort of 160 people with schizophrenia who were recruited in 2016 for a four month (16 weeks) prospective observational study. It then seems that case records were augmented with telephonic collection of self-report data regarding the previous 8 years (i.e., recall/retrospective and cross-sectional data) where the individual/ family member could be contacted. Were there other forms of contact? - were there in-person/ clinical interviews? The time periods of record review (presumably some time in 2024?) and telephonic contact should be reported.

- Results

While table 1 and 2 provide baseline characteristics of the 160 people, this article is concerned with the cohort at 8 years after initial recruitment. Therefore, it would be more accurate to report on the baseline characteristics of the current study sample (i.e., n=147), and could include all the variables analysed later in tables 4 and 5. However, note that of the 160, n=13 had inadequate records and therefore excluded from the study because no data (so presumably disengaged, but time to disengagement and possible reasons are unknown). The baseline characteristics of these 13 could be reported for completeness, but they should not be included in the 8 year retrospective cohort of 147.

The sentence "At the 8th year of follow-up, we had contact with 147 (91.9%) of the original 160 subjects and, or

their family caregivers, mostly by telephone interviews, with information supplemented from the case record.." needs clarifying. What does "contact" mean? Does it mean that all 147 were interviewed, with 90 interviewed telephonically and 57 interviewed in person? Or that case records were available for 147of the original 160 ...? From the methods, it appears that case records were used primarily and supplemented with telephonic interviews in 90 people ... please clarify.

Table 3 ... suggest reporting these data prior to Kaplan-Meier survival curve. ALso, the table needs some revision ... would be easier to read if all variables are reported for the whole sample (n=147), accounting for missing data, or if not applicable (e.g., for the cause of death ... although this could also be reported in the text). So, where information is only available for 90 participants (are these the 90 who were contacted telephonically?), missing data for 57 should be noted.

Kaplan-Meier curve ... is this for the sample of 147 or 160?

Tables 4 and 5 ... were the variables for all 160, or only the 147 whose data were available at 8 years, and for whom time to disengagement was known?

Other - I am wondering if there was any significant association with relapse / number of relapses over the 8 years and disengagement?

Discussion

The high rate of disengagement suggests that the population in question may not perceive the western medical model of care as necessary ... what are the perceived needs of people with schizophrenia in Nigeria? Should we be addressing these in our efforts to retain people in care?

I am concerned about some overstatement of the facts. For example, the word "many" in "many returning to the earlier path of care (i.e, faith-based)" suggests at least half, when 38/147 (25%) sought help from alternative healers and only 3/147 (2%) continued with both alternative and western healers.

Reviewer #2: Overall comment

The study examined time to treatment discontinuation in a cohort of patients with schizophrenia in Sub-Saharan African (SSA) region. Non-treatment and discontinuation of treatment is a well-known major problem in communities in this regions. But treatment discontinuation is a widespread problem in the world, more acutely noted in areas with low economic and health service resources and in cultures with alternative models of illness and treatment. This study carried the potential to carry the issue beyond identifying treatment discontinuation to describe and discuss factors that are potentially amenable to change and intervention to improve treatment access rates. It appears the authors had the data but their limited focus on time to discontinuation limited the scope of the study which did not add to the knowledge on the issue of treatment continuation. Other important factors like the socio-economic and cultural reality, and the access and quality of services available that can push the patients away from seeking treatment need greater attention in the discussion. Such factors are a barrier to patients seeking treatment and can be addressed directly by the service systems

Specific Comments

Methods Section

The role of the family/social network in the whole process of treatment seeking and continuation need description.

The method of measuring the DUP measured (timing the onset of illness, Indicators of onset) needs description.

A definition of improvement was given but this factor was not relevant for follow up as this issue was not studied.

Pg 7 and 9. The mode of follow up needs a bit more clarity regarding the mode of contact, the nature of data collected, and assessments done. The different types of contact need description. It can have a bearing on the discussion on how a service can access the population of the type featuring in the study over long term.

Considering interval to discontinuation in whole years may have resulted in loss of information. Graduated intervals like 3,6,12, and 12+ months may have been useful in differentiating patients disengaging very early and their characteristics compared to those who stopped after at least an year of treatment.

Results section

(Page 10) When interpreting data based on statistical analysis, the degree of non-significance does not give the scope to interpret "tendencies". Interpretation of the relationships between dependent and independent variables need to adhere to the level of significance set for data interpretation.

Discussion section

The study explored only the time to disengagement but not variables related to disengagement event per se. It did not identify any independent variables associating with time to disengagement. The references to other works on the issues should be place against the study's observations. The discussion addressed many issues not directly related to the data and results of this study.

The absence of any association of any factor with disengagement is an interesting finding that can receive a good amount of discussion. Issues like access to the follow up appointments, the role of faith/religious healing moving patients away from formal care services, the local socio-cultural understanding and dealing with mental health issues could have been brought up in discussion with comments on how such issues can be addressed will add a translational perspectice to dealing with the well-known reality that utilisation of MH services is low in the study region. Such an attempt could increase the impact of this work.

Presentation

Some work on the language could help improve the quality and the clarity of presentation.

Reviewer #3: This is well written manuscript on an important topic. I appreciate the authors for this research.

I have a few comments

1. The title of the study is clear

2. The abstract is comprehensive.

3. The review of literature is adequate

4. Methodology and definitions of the key variables are accurate. I acknowledge the thorough approach to collecting follow-up data, especially given the challenges of long-term studies. However, some participants were unreachable, and certain case notes were missing. It would be helpful if the manuscript explained how missing data were handled, such as whether imputation techniques or sensitivity analyses were used to reduce bias.

5. Can the authors briefly explain how the sample size of 160 participants was determined? Specifically, whether power analyses were conducted or expected disengagement rates were considered to ensure the study is adequately powered?

6. The logistic regression analysis offers valuable insights, but no significant predictors of disengagement were found. This might be due to limited statistical power. A brief discussion about the study's power to detect associations or consider additional predictors, like socio-economic status or medication adherence, could enhance the analysis.

7. The Kaplan-Meier method is appropriate for analyzing time-to-event data like patient disengagement. It would be beneficial for the authors to clarify how they managed cases of censoring, especially for participants who were unreachable or had missing case notes.

8. The authors have cited two factors as limitations. However, this is a naturalistic study . It is not uncommon to miss case notes or not be able to talk to the participants. As such, I would recommend that the authors discuss this as a natural phenomena and not project is as a limitation. The authors may also find the paper by Lee et al 2021, useful. https://pmc.ncbi.nlm.nih.gov/articles/PMC8168830/

**Do you want your identity to be public for this peer review?** For information about this choice, including consent withdrawal, please see our Privacy Policy

Reviewer #1: **Yes: ** Lesley Robertson

Reviewer #2: No

Reviewer #3: No

---

## [Decision Letter · Decision Letter 1]

13 Jan 2025

Disengagement from Treatment and its Socio-demographic and Clinical Predictors among Patients with Incident Schizophrenia in a Nigerian Psychiatric Hospital: 8-Year Naturalistic Follow-up Analysis

PMEN-D-24-00308R1

Dear Dr. Onu,

We are pleased to inform you that your manuscript 'Disengagement from Treatment and its Socio-demographic and Clinical Predictors among Patients with Incident Schizophrenia in a Nigerian Psychiatric Hospital: 8-Year Naturalistic Follow-up Analysis' has been provisionally accepted for publication in PLOS Mental Health.

Best regards,

Karli Montague-Cardoso

Executive Editor

PLOS Mental Health

Reviewer Comments (if any, and for reference):

Reviewer's Responses to Questions

Comments to the Author

Reviewer #1: All comments have been addressed

Reviewer #2: All comments have been addressed

publication criteria?

Reviewer #1: Yes

Reviewer #2: Yes

3. Has the statistical analysis been performed appropriately and rigorously?

Reviewer #1: Yes

Reviewer #2: I don't know

4. Have the authors made all data underlying the findings in their manuscript fully available (please refer to the Data Availability Statement at the start of the manuscript PDF file)?

Reviewer #1: Yes

Reviewer #2: Yes

5. Is the manuscript presented in an intelligible fashion and written in standard English?

Reviewer #1: Yes

Reviewer #2: Yes

Reviewer #1: Thank you for your detailed responses to my comments - the article is great!

Reviewer #2: No comments

**Do you want your identity to be public for this peer review?** For information about this choice, including consent withdrawal, please see our Privacy Policy

Reviewer #1: **Yes: ** Lesley Robertson

Reviewer #2: No
